# OpenReview forum: "@grokSet: multi-party Human-LLM interactions in social media"
_ICML.cc/2026/Conference — Submitted to ICML 2026_

### Official Review · Reviewer_qH5R · 2026-03-11

**Soundness:** 3
**Presentation:** 3
**Significance:** 3
**Originality:** 3
**Overall Recommendation:** 4
**Confidence:** 2

**Summary:**

This paper introduces @GROKSET, a dataset of over 1M tweets and 182K conversation threads involving the Grok LLM on X. Unlike prior datasets based on private chatbot logs, it captures public, multi-party human–LLM interactions on social media. The paper examines the concept of LLMs acting as participants in public discourse. Using topic modeling, engagement analysis, and safety evaluation, the authors find that users often invoke the model as an arbiter in political debates, that LLM responses receive less engagement than human replies, and that safety alignment can be bypassed through persona adoption and tone mirroring.

**Compliance With Llm Reviewing Policy:**

Affirmed.

**Final Justification:**

The paper presents a novel and timely dataset of public human–LLM interactions on social media, with a sound empirical methodology and clear presentation. The authors addressed my questions well. I would recommend more validation since inter-annotator agreement on several key dimensions is low. Therefore, I will maintain my score.

**Key Questions For Authors:**

1. The dataset focuses on Grok interactions on X. How generalizable are the findings to other platforms or LLM systems? Clarification could affect how broadly the results apply.

2. Could missing or deleted tweets introduce bias (e.g., underestimating toxicity or engagement patterns)? More detail on this would help interpret the results.

3. The paper relies on LLM-based annotation and automated toxicity detection. Was any validation with human annotation performed to ensure reliability?

**Limitations:**

yes

**Strengths And Weaknesses:**

Soundness:
The dataset construction, methodology, and analysis pipeline are described clearly and follow a reasonable empirical design. However, some claims, such as framing LLMs as “arbiters” in debates, rely partly on qualitative interpretation and would benefit from clearer operational definitions and stronger quantitative justification.

Presentation:
The paper is generally well organized and clearly written. The dataset construction, methodology, and analysis are presented in a structured manner, and figures and examples help illustrate the main findings.

Significance:
Studying LLM behavior in real-world social media environments is important, as it differs from traditional private chatbot settings. The dataset could be a useful resource for research on human–AI interaction, online discourse, and AI safety. The focus on a single platform (X) and model (Grok) may limit the generalizability of the results.

Originality:
The main novelty lies in introducing a large-scale dataset of multi-party human–LLM interactions on social media, which fills a gap in existing resources. The analytical techniques used are largely standard, and several observations (e.g., jailbreak through role-play or adversarial prompting) largely confirm patterns already documented in prior alignment research.

---

> ### Author Rebuttal · Authors · 2026-03-31
>
> ## Generalizability to other platforms and LLM systems (Q1)
> We agree that our findings are specific to Grok's persona and X's platform culture, and will expand the Limitations section accordingly:
> > _Our findings are inherently tied to Grok's specific persona and X's platform culture; additionally, our sampling likely over-represents a demographic that intentionally invokes the LLM._
>
> We would note, however, that this specificity is precisely what makes @GROKSET an ecologically valid record of AI behavior in a real deployment. Grok's persona likely has an impact, but the structural conditions that enable it (multi-party threading, public visibility, adversarial social pressure) are platform-level properties that any LLM embedded in X's reply graph would face. X currently represents the largest live experiment of this kind, categorically different from private chatbot interactions and largely absent from prior datasets; the patterns we document are therefore empirical baselines against which future deployments, across both platforms and models, can be measured.
>
> ## Deleted tweets (Q2)
> @GROKSET is released in a dehydrated (Tweet-ID-only) format, which means tweets that have been deleted or made private at the time of rehydration are not recoverable. Therefore, this introduces a systematic rather than random bias: deleted content on X disproportionately includes posts removed by automated moderation or platform safety systems: meaning the most egregious safety failures are precisely the ones most likely to be absent from the corpus. As we state in Section 8 (L431-L438), our reported toxicity rate should therefore be interpreted as a conservative lower bound. Our toxicity analysis in Section 7 is accordingly framed not as "Grok is toxic X% of the time" but rather as an investigation into what conversational conditions are associated with safety failures: a causal framing that is robust to this censoring, since the surviving examples are sufficient to identify the relevant triggers (persona adoption, tone mirroring). We will add a sentence in Section 6 to make the directional implication for the engagement analysis equally explicit.
>
>
> ## Human annotation validation for LLM-based annotations. (Q3)
> Our annotation pipeline has distinct components, each validated through methods appropriate to its nature.
> For toxicity detection, we use Detoxify [1], a model trained on the Jigsaw Toxic Comment Classification Challenge and extensively validated against human annotations prior to our study, achieving an AUC of 98.64%. This external validation record means Detoxify's reliability does not depend on in-study human annotation; it enters our pipeline as an already-validated instrument.
> For the qualitative annotation layer (Gemini 2.0 Flash), we conducted an in-study human validation. Three expert annotators independently labeled 65 discussions across the core annotation dimensions. We measured inter-annotator agreement among human experts using Fleiss' Kappa, and computed Gemini's agreement against the human majority-vote consensus.
>
> | Annotation | Expert Agreement | Gemini Accuracy vs. Consensus |
> |:--:|:--:|:--:|
> | `is_discussion` | 0.659 | 98.5% |
> | `assistant_stance` | 0.605 | 95.4% |
> | `bias_language` | 0.381 | 90.8% |
> | `user_response_type` | 0.466 | 81.5% |
> | `discussion_intensity` | -- | 0.822 (Pearson) |
>
> Human experts show substantial agreement on the primary classification task ($\kappa = 0.659$), confirming that the annotation schema is well-defined. For more subjective dimensions such as bias language and user response type, lower inter-annotator agreement is expected, and is consistent with established findings in NLP annotation research [2]. Crucially, Gemini tracks the human consensus closely across all dimensions, including the more subjective ones, suggesting that its outputs are not introducing systematic distortion beyond the irreducible ambiguity that human annotators themselves face. We will surface both the Detoxify external validation figures and the Gemini human-agreement results more prominently in the revised manuscript.
>
> - [1]: Laura Hanu and Unitary team. 2020. Detoxify. github.com/unitaryai/detoxify
> - [2]: Ellie Pavlick and Tom Kwiatkowski. 2019. Inherent Disagreements in Human Textual Inferences. Transactions of the Association for Computational Linguistics, 7:677–694.

---

> > ### Author Rebuttal · Reviewer_qH5R · 2026-04-06
> >
> > I thank the authors for the detailed rebuttal. The inter-annotator agreement results and Gemini accuracy figures are reassuring. I will keep my score.

---

### Official Review · Reviewer_JgMr · 2026-03-12

**Soundness:** 2
**Presentation:** 2
**Significance:** 2
**Originality:** 2
**Overall Recommendation:** 2
**Confidence:** 4

**Summary:**

This paper introduces @GrokSet, a large-scale dataset of over 1 million tweets involving the @grok LLM on X. They describe the dataset, illustrating that rather than serving as a general assistant, Grok is frequently invoked as an authoritative arbiter in high-stakes, polarizing political debates. The model functions as a low-status utility, receiving significantly less social validation (likes, replies) than human peers. The authors report that users bypass safety filters not through complex jailbreaks, but through simple persona adoption and tone mirroring.

**Compliance With Llm Reviewing Policy:**

Affirmed.

**Key Questions For Authors:**

1. What is the machine learning contribution beyond dataset release? The paper makes a clear resource contribution, but what specifically is the ML problem that this dataset helps solve?

2. What kinds of tasks do the authors envision this dataset enabling that existing human–AI or social media datasets cannot support? A clearer benchmark framing would help readers understand why this resource matters for the ML community.

3. What preprocessing and filtering decisions were made during dataset construction? The paper would benefit from more detail on collection criteria, deduplication, thread reconstruction, language filtering, bot handling, and exclusion rules.

4. How should users of the dataset think about sampling bias? Public mentions of Grok may overrepresent certain types of users, topics, or interaction styles. A more explicit discussion of these biases would strengthen the paper.

5. What are the main ethical constraints on reuse? Since the dataset is drawn from public but potentially sensitive conversational contexts, what guidance do the authors provide regarding privacy, re-identification, or harmful downstream use?

6. Can the authors provide stronger baseline analyses or benchmark tasks? Even a small set of predictive or inferential tasks would help demonstrate the analytical value of the dataset beyond simple description.

**Limitations:**

The paper should discuss the implications of working with public social media conversations involving an AI assistant. Even if the data are publicly accessible, the release of large-scale reconstructed interaction threads can increase risks of re-identification, context collapse, and downstream misuse. In addition, descriptive analyses of user behavior in this setting may be overinterpreted as broader claims about human–LLM interaction when they may instead reflect the specific culture of X and the specific persona of Grok. A stronger limitations section would therefore do three things more explicitly: first, narrow the claims about generalizability; second, discuss platform and sampling bias; and third, address privacy and misuse risks associated with releasing and analyzing large-scale public human–AI conversations.

**Strengths And Weaknesses:**

Strengths

Novel and timely dataset
The paper introduces a large-scale dataset of public interactions with a deployed LLM in a real social media environment. This is timely and potentially valuable, especially because most existing human–AI interaction datasets are either synthetic, lab-based, or platform-restricted.

Scale and ecological validity
The dataset is large and drawn from naturally occurring conversations on X. This gives it clear value as a record of how users actually engage with Grok in a public, multi-party setting.

Potential utility for future research
The resource could support work in computational social science, NLP, and human–AI communication. It may be especially useful for studying public prompting behavior, conversational norms around LLM use, and the social dynamics of AI-mediated interaction in open online environments.

Weaknesses

Limited methodological contribution
The main contribution is the construction and release of a dataset, accompanied by descriptive analysis. The paper does not introduce a new machine learning method, modeling framework, or evaluation paradigm. This limits its contribution for a venue such as ICML, where stronger methodological novelty is usually expected.

Lack of ML-specific technical insight
The analysis appears to rely primarily on existing tools and descriptive summaries rather than on new modeling ideas or technically substantial empirical benchmarks. As a result, the paper does not yet show what new machine learning questions this dataset uniquely enables.

The empirical analysis remains mostly descriptive
The current analysis is useful as dataset characterization, but it remains fairly preliminary. The paper would be stronger if it moved beyond summary statistics and topic description toward more substantive analytical or predictive tasks.

Limited generalizability beyond Grok on X
The dataset is specific to one model, one platform, and one interaction environment. It is therefore unclear how far either the descriptive findings or the broader claims can extend to other LLMs, other social media systems, or private chat-based interaction settings.

Weak fit for ICML
The contribution seems better aligned with NLP, computational social science, or human–computer interaction venues than with a machine learning conference centered on methodological innovation.

---

> ### Author Rebuttal · Authors · 2026-03-31
>
> We appreciate the reviewer's acknowledgment of the dataset's novelty and ecological validity, and address the core concerns below.
> ## ML Contribution, Benchmark Tasks, and Dataset Distinctiveness (Q1, Q2, Q6)
> The core problem class @GROKSET uniquely enables is context-dependent inference in multi-party conversations: predicting, classifying or explaining model behavior from the accumulating social and conversational context in which they occur. This differs fundamentally from existing resources. LMSYS-Chat-1M and WildChat are private dyads: one user, one model, no audience. @GROKSET introduces multiple users with conflicting goals, adversarial pressure, branching thread topology, and live engagement signals. Existing datasets cannot pose this learning problem.
>
> As a concrete benchmark, we evaluate *toxicity forecasting*: _"given N turns of a multi-party thread, can a model predict whether the LLM's next response will exceed toxicity thresholds?"_ Using a subsample of 4,365 threads (80/20 split), we benchmarked:
>
> | Method | F1 Score | Recall |
> |---|---|---|
> | DistilBERT | 0.61 | 0.71 |
> | LogReg (TF-IDF) | 0.56 | 0.60 |
> | SVM (TF-IDF) | 0.44 | 0.31 |
>
> Interestingly performance increases with thread depth (1–3 turns: F1=0.44; 4–9 turns: F1=0.59; 10+ turns: F1=0.81). This depth-stratified result is itself a finding: safety classifiers trained on isolated turns miss the signal that accumulates over multi-party exchanges, a signal that only exists in @GROKSET.
>
> But toxicity forecasting is just one entry point into the broader problem class. Engagement prediction may recast the same structure as a supervised regression task, predicting likes, replies, etc... Opinion shift modeling would treat multi-user political threads as a testbed for causal inference on LLM-mediated persuasion, tracking stance changes under model intervention. Alignment stress-testing could use the adversarial pressure of public discourse to probe safety mechanisms in ways synthetic jailbreak datasets cannot replicate. What unifies all of these is the same requirement: a dataset with multi-party structure, contextual depth, and social grounding. That is what @GROKSET provides and what no prior resource does.
> ## Data Curation and Preprocessing (Q3)
> Our pipeline (Appendix B) proceeds as follows: (1) seed sampling of @grok replies in six-hour blocks across Mar–Oct 2025 (≤300 conversations/block); (2) recursive thread reconstruction via `in_reply_to_id`; (3) termination if no Grok message appears within 15 consecutive parents or the thread exceeds 150 turns; (4) deduplication by unique `tweet_id`. Language filtering was intentionally omitted, as threads are frequently multilingual (Fig. 2b). We will clarify this further in Appendix B.
>
> ## Generalizability and Platform Bias (Q4)
> We agree that our findings are specific to Grok's persona and X's platform culture, and will expand the Limitations section to state this more explicitly as follows:
> > _"Our findings are inherently tied to Grok's specific persona and X's platform culture; additionally, our sampling likely over-represents a demographic that intentionally invokes the LLM."_
>
> ## Ethical Safeguards and Dehydrated Release (Q5)
> The risk of reidentification or context collapse is real, but we need to be precise: the tweets in @GROKSET were posted publicly and are already indexable by search engines. Any researcher who rehydrates via the X API will receive original usernames and user IDs by design. This is an unavoidable property of public social media datasets and standard practice in the field (e.g., RedCaps [1]). Dehydrated release is the strongest privacy protection available within platform constraints: if a user deletes a tweet after our collection window, rehydration fails automatically, honoring their right to be forgotten under GDPR without requiring us to police individual deletions. Legal compliance responsibility rests with the rehydrating researcher, as is conventional. This is mentioned in Section 8, but will be made more clear in the camera-ready.
>
> **Usage guidelines.** We will include a datasheet [2] and explicit permitted/prohibited use documentation in the repository. @GROKSET is intended for research on human–LLM interaction, safety evaluation, and computational social science. Uses that train systems to replicate alignment bypasses, build individual behavioral profiles, or target or identify specific users are explicitly out of scope.
>
> - [1]: Desai et al. (2021). RedCaps: Web-curated image-text data created by the people, for the people. In Proceedings of the Neural Information Processing Systems.
> - [2]: Gebru et al. (2021). Datasheets for datasets. Commun. ACM 64, 12 (December 2021), 86–92.

---

### Official Review · Reviewer_Jb9w · 2026-03-13

**Soundness:** 2
**Presentation:** 3
**Significance:** 2
**Originality:** 3
**Overall Recommendation:** 4
**Confidence:** 5

**Summary:**

# RQ
* How do LLMs behave and interact in unconstrained, multi-party public social environments compared to private, 1-on-1 chats?
* What specific roles do users assign to LLMs when they are embedded in public social media platforms like X?
* How robust are model safety mechanisms when subjected to the adversarial and performative pressures of public discourse?

# Method
* network modeling (centrality, reciprocity, transitivity) for multi-party interactions
* Linear Mixed-Effects Models to analyze social engagement (likes, replies)
* BERTopic for automated topic modeling
* Detoxify for toxicity detection
* LLM-as-a-judge framework for qualitative analysis of conversational dynamics.

# Contribution
* @GROKSET, a large-scale corpus of over 1 million tweets (182k conversations) involving the Grok LLM on X over a seven-month period in 2025.
* LLM outputs receive significantly less social validation
* public users easily bypass safety filters through simple tone mirroring and persona adoption rather than complex technical jailbreaks.

**Compliance With Llm Reviewing Policy:**

Affirmed.

**Final Justification:**

Q1 was partially resolved (3 annotators, on 65 labels). While it does provide some good info, it's too small of a sample to really trust the result.
Q3 was resolved.

**Key Questions For Authors:**

1. Human-annotated baseline & inter-annotator agreement score (e.g., Cohen’s Kappa) to validate the Gemini 2.0 Flash annotations
2. Dataset, can you try alternative strategies for sharing aggregated, anonymized semantic features (e.g., topic distributions or embeddings)?
3. Benchmarking the Grok's "engagement gap" against other known automated or bot accounts on X, rather than exclusively comparing it to human peers? or use botometer to classify accounts as bots and see their engagement ratios?


( if all Q1 and Q3 are done, will bump to accept)

**Limitations:**

yes

**Strengths And Weaknesses:**

# Soundness

## Strength
- Linear Mixed-Effects Model (LMM) to analyze engagement, correctly isolating the effect of the author's identity (Grok vs. Human)

## Weakness
- LLM-as-a-judge (Gemini 2.0 Flash) to label behaviors of trolling, intensity & bias (no human annotation on subset etc)
- @Grokset is dehydrated Tweet IDs (while they can't release the full dataset due to X's terms and conditions), cases where tweets are deleted & accessibility issues

# Presentation
## Strength
- honest about the dataset's survivorship bias (toxic ones are removed by moderators / automated moderation)
- clear, grounded, effective visualization
## Weakness
- "Engagement Gap" as a major, titled discovery rather than an expected baseline behavior (humans naturally do not "like" or validate bot replies as much as human replies)

# Significance
## Strength
- timely, very important, LMs are integrated into public social platforms increasingly
- successfully advances our empirical understanding of "shallow alignment" in the wild
## Weakness
- engagement gap (expected baseline of human behavior, not a novel capability or vulnerability discovery)
- idea of the dataset could inspire future research, the actual dataset's utility is decreased significantly by its dehydrated release format (Tweet IDs only)
- heavy Western/English-centric demographic skew of a single platform

# Originality
## strength
- novel context
## weakness
- "Shallow Alignment" via persona adoption (e.g., "DAN" prompts) is already heavily documented in adversarial literature

---

> ### Author Rebuttal · Authors · 2026-03-31
>
> We thank the reviewer for the careful, technically rigorous assessment.
>
> ## Validating LLM annotations (Q1)
> Yes, we performed a rigorous human validation study to ensure the reliability of the Gemini-based automatic annotations. We tasked three expert annotators to independently label a sample of 65 discussions, covering one of the core annotations of our study.
> We evaluated two aspects. First, we measured inter-annotator agreement among the human experts using Fleiss’ Kappa. Second, we computed the agreement between Gemini and the human consensus, defined as the majority vote.
>
> | Annotation Metric | Expert Agreement ($\kappa$) | Gemini Accuracy (vs. Human Consensus) |
> | :--- | :--- | :--- |
> | `is_discussion` | 0.659  | **98.5%** |
> | `assistant_stance` | 0.605 | **95.4%** |
> | `bias_language` | 0.381 | **90.8%** |
> | `user_response_type` | 0.466 | **81.5%** |
> | `discussion_intensity` | — | **0.822** (pearson) |
>
> Human annotators show substantial agreement on the primary task (`is_discussion`, $\kappa = 0.659$), indicating that the annotation schema is reliable. Gemini closely matches this consensus, achieving 98.5% accuracy on discussion identification and 95.4% on assistant stance.
> For more subjective categories such as bias language and user response type, inter-annotator agreement is lower, which is consistent with prior work in NLP [1]. Despite this, Gemini maintains strong alignment with the majority vote, reaching 90.8% accuracy on bias language.
>
> **Toxicity Detection:** Toxicity in our pipeline is measured using Detoxify (Hanu and Unitary team, 2020). This model is trained on the Jigsaw Toxic Comment Classification Challenge dataset and has been extensively validated against human annotations, achieving an AUC of 98.64 percent.
>
> ## Dataset sharing (Q2)
> While X's Terms of Service and GDPR constraints prevent us from releasing full tweet texts or user handles, we do release derived, non-identifiable features alongside the tweet IDs: topic cluster assignments (BERTopic), Gemini-generated discussion labels, toxicity scores, engagement metadata, and thread-level network statistics. Rehydration of the full 1M+ tweet dataset is straightforward using our provided code, with an estimated one-time API cost of approximately $150, accessible to most academic research groups. We will make this cost estimate and the rehydration pipeline more prominent in the data release documentation.
>
>
> ## Engagement gap in bot accounts (Q3)
> We agree that comparing Grok's engagement to known automated accounts provides a more informative baseline than a human-only comparison. We queried Botometer for all users appearing in at least two threads who created their accounts before November 2023 (n = 28,612 eligible users).
>
> *Correlation between bot score and average engagement:*
>
> | Metric | Pearson r | Spearman ρ |
> |---|---|---|
> | Avg. likes | −0.075 | −0.183 |
> | Avg. replies | −0.060 | −0.148 |
>
> For the high-confidence bot subset (bot score > 0.8; n = 277), average engagement is 60.4 likes and 7.6 replies, compared to 222.6 likes and 14.5 replies for the remaining eligible users, roughly −72.9% likes and −47.7% replies. Grok's own average (38.8 likes, 2.6 replies across 450,178 tweets) falls *below* even this likely-bot cohort, suggesting that the engagement gap may be correlated with automated-account dynamics.
>
> We acknowledge Botometer's limitations: it is based on pre-LLM-era signal (data through November 2023) and cannot be updated due to X's current API restrictions, and the high-confidence bot subset is small relative to the full population. We therefore present this analysis as corroborating evidence rather than a definitive benchmark, and note that no modern equivalent tool currently exists. We will add this analysis and its caveats to the paper.
>
> - [1]: Ellie Pavlick and Tom Kwiatkowski. 2019. Inherent Disagreements in Human Textual Inferences. Transactions of the Association for Computational Linguistics, 7:677–694.

---

> > ### Author Rebuttal · Reviewer_Jb9w · 2026-04-03
> >
> > Score has been updated.

---

### Official Review · Reviewer_4awn · 2026-03-13

**Soundness:** 3
**Presentation:** 4
**Significance:** 3
**Originality:** 4
**Overall Recommendation:** 5
**Confidence:** 4

**Summary:**

This paper presents a dataset of over 1 million tweets from conversations involving the Grok language model on the X (Twitter) platform, featuring multi-party conversations involving LLM assistants. Authors draw three key findings from the collected data: the langauge model is frequently invoked as an authoritative arbiter in polarizing political debates rather than as a general assistant in the X platform; it receives significantly less social engagement (likes, replies) than human participants in the same threads; and its safety alignment is easily bypassed not through sophisticated jailbreaks but through simple persona adoption and tone mirroring.

**Compliance With Llm Reviewing Policy:**

Affirmed.

**Final Justification:**

**Post-Rebuttal**:
Authors have provided additional results and explanations that help resolve the said weaknesses and questions.
Accordingly updating my evaluation of the submission.

While it is true that there are additional/further analyses possible with the presented dataset and methodological contributions, but this is nature of all contributions that newly propose, process, and provide datasets that advance the field. My review reflects this nature of the presented work, as a dataset contribution that is novel and appropriate in advancing our understanding of how human users interact with LLM-based systems on social media.

**Key Questions For Authors:**

* Have you double checked compliance with X's Terms of Service in releasing the dehydrated dataset?

**Limitations:**

Yes.

**Strengths And Weaknesses:**

Strengths:
* The presented dataset is an interesting, valuable collection surfacing a new type of human-AI interaction on social media. The proposition of taking the Grok-involved tweet threads, in the wild, to understand multi-user interactions, mediated by and in relations to LLM systems, is well motivated.

Weaknesses/Suggestions:
* Analysis on reciprocity and transitivity across tweet threads is interesting. Further analysis such as tying the thematic analysis against these trends, e.g. are there distinct themes correlated with high recipriocity and transitivity? Do multiple users agree/disagree to the same Grok responses, or does Grok respone with similar answers (say with the same political stance) across users in the same thematic context?
* Analysis in Section 6 on the engagement gap yield rather intuitive results: engagement metrics for both human users and the LLM decrease over thread depth, possibly conflated by the recommendation algorithm on the X platform and how users interact with the platform. While the authors propose an interesting hypothesis explaining the user differentiation of LLM responses (lower engagement that human tweets), it would be intersting to see an analysis validating the claim. For instance, do we see different human-Grok engagement gaps for political-charged threads versus information-centric threads (e.g. "Identifying Movie Scenes") where the neutral, information-delivering quality of LLMs could be more relevant in human treatment of models as social peers?

---

**Post-Rebuttal**:
Authors have provided additional results and explanations that help resolve the said weaknesses and questions.
Accordingly updating my evaluation of the submission.

---

> ### Author Rebuttal · Authors · 2026-03-31
>
> ## Transitivity in High-stakes topics (Q1)
> We thank the reviewer for this constructive suggestion. To address it, we segmented conversations by topic cluster, categorizing our top 20 topics as either *High-Stakes* or *Neutral*. The results reveal meaningful structural differences between the two categories.
> High-Stakes threads show substantially higher network cohesion than Neutral ones, with average transitivity 25.9% higher (0.313 vs. 0.248) and average reciprocity 66.1% higher (0.038 vs. 0.023), indicating tighter, more contested multi-party exchanges in politically-charged contexts. Discussion intensity follows the same direction (1.878 vs. 1.756 on average).
>
> | Topic | Category | Transitivity | Reciprocity | Avg Intensity |
> |---|---|---|---|---|
> | Zionism and Antisemitism | High-Stakes | 0.411 | 0.045 | 1.941 |
> | Biblical Interpretation | High-Stakes | 0.383 | 0.059 | 1.766 |
> | Islamic Scripture | High-Stakes | 0.359 | 0.071 | 1.864 |
> | Racial Disparities in Crime | High-Stakes | 0.354 | 0.056 | 1.941 |
> | Kirk Shooting | High-Stakes | 0.337 | 0.034 | 1.333 |
> | Vaccine Safety | High-Stakes | 0.324 | 0.051 | 1.944 |
> | Foreign Aid | High-Stakes | 0.318 | 0.020 | 1.804 |
> | Theological & Scientific Debate | Neutral | 0.311 | 0.036 | 1.829 |
> | AI Image Verification | Neutral | 0.307 | 0.027 | 1.791 |
> | AGI Sentience | Neutral | 0.305 | 0.020 | 1.746 |
> | Turkish Kurdish Relations | High-Stakes | 0.304 | 0.028 | 1.802 |
> | Conversation Initiation | Neutral | 0.300 | 0.024 | 1.628 |
> | Ukraine Russia Conflict | High-Stakes | 0.291 | 0.035 | 1.836 |
> | Elon Musk and Ventures | High-Stakes | 0.253 | 0.010 | 1.794 |
> | Nigerian Political Rivalry | High-Stakes | 0.252 | 0.034 | 1.815 |
> | Fact Checking Politics | High-Stakes | 0.229 | 0.026 | 1.857 |
> | Canadian Economic Policy | Neutral | 0.218 | 0.030 | 1.795 |
> | Flirtatious Role-Playing | Neutral | 0.375 | 0.032 | 1.871 |
> | Bitcoin and Gold | Neutral | 0.120 | 0.010 | 1.550 |
> | Identifying Adult Performers | Neutral | 0.099 | 0.004 | 1.526 |
> | **High-Stakes avg** | | **0.313** | **0.038** | **1.878** |
> | **Neutral avg** | | **0.248** | **0.023** | **1.756** |
>
> We will incorporate this analysis into the camera-ready revision.
>
> ## Engagement in High stakes topics (Q2)
> We want to take the opportunity to clarify that the decrease in engagement over thread depth was not our main finding but rather a control condition used to verify the primary result (that Grok receives meaningfully less engagement than human participants) and we will make this distinction more explicit in the revision.
> More importantly, we ran the analysis the reviewer suggests, and the results strongly validate the engagement gap finding. We segmented threads into High-Stakes and Neutral categories (as defined in Q1) and measured the reply gap: the percentage difference in average replies received by humans versus Grok. In High-Stakes threads, humans received **175% more replies** than Grok on average (4.98 vs. 1.81). In Neutral threads, this gap narrows substantially to just **54%** (2.52 vs. 1.63), more than three times smaller. The same pattern holds for likes.
> | Category | Author | Avg. Replies | Avg. Likes | Reply Gap (%) | Like Gap (%) |
> |---|---|---|---|---|---|
> | **High-Stakes** | Grok | 1.81 | 19.10 | **175%** | **144%** |
> | | Human | 4.98 | 46.65 | | |
> | **Neutral** | Grok | 1.63 | 11.24 | **54%** | **45%** |
> | | Human | 2.52 | 16.34 | | |
>
> Further granularity reveals that in extreme high-stakes topics such as *Ukraine-Russia Conflict* or *Charlie Kirk Shooting*, the human reply advantage exceeds 290% and 630% respectively. In contrast, neutral topics like *AI Image Verification* or *AGI Sentience* show much narrower gaps (18% and 66%), aligning with our finding that neutral contexts facilitate a closer treatment of LLMs as social peers.
>
>
> ## ToS Compliance
> We confirm compliance with X's Developer Agreement and Policy. Sharing only Tweet IDs and derived annotations is the standard and explicitly permitted method for academic dataset sharing under X's content redistribution policy. Redistribution of raw tweet content is prohibited, but sharing of Tweet IDs is allowed. Researchers rehydrating the dataset will only retrieve content still publicly available at query time, thereby respecting user deletions and privacy settings. This approach is consistent with standard practice across academic datasets derived from the platform.

---

> > ### Author Rebuttal · Reviewer_4awn · 2026-04-04
> >
> > Thank you for the responses and extra explanations. The expanded analyses and discussions (Q1-Q2) will be great to be included in the manuscript for the readers; ToS compliance should be made clear with respect to the dataset release.

---

### Decision · Program_Chairs · 2026-04-30

**Decision:**

Reject

**Comment:**

This paper presents @GROKSET, a large-scale corpus of more than one million tweets and roughly 183 thousand threads involving Grok on X, and use it to support the following claims: (1) users often invoke the model as an arbiter in politically and socially charged debates. (2) model outputs receive substantially less engagement than human replies within the same threads. (3) Some safety failures arise through relatively simple persona adoption and tone mirroring rather than more elaborate jailbreaks. The paper’s main contribution is the dataset itself together with an initial empirical characterization of this interaction setting.

I have some methodological concerns around  the findings being  tied quite closely to one platform, one deployed model, and one platform culture, so the claims are very limited to this scope. The annotation validation is helpful, but the human validation set is still fairly small, and agreement is visibly lower on more subjective dimensions. The dehydrated release format is understandable and appropriate under the platform constraints, but it does reduce ease of reuse and introduces unavoidable survivorship effects. The authors do acknowledge that deleted or moderated content likely makes the reported toxicity rates conservative lower bounds, which is the right framing, but this still limits how broadly some conclusions should be interpreted.

Most importantly, agreeing with reviewer JgMr (and disagreeing with the other reviewers), I do not believe this paper is a good fit for ICML, which is primarily a machine learning conference. Even taking the rebuttal into account, the paper’s core contribution remains a large observational dataset and a set of descriptive analyses of Grok interactions on X rather than a clear machine learning advance. The rebuttal does help by articulating a more concrete ML angle around context-dependent inference in multi-party public threads, and the toxicity-forecasting example makes the dataset look more benchmark-like than the submission initially suggested. But that framing still feels somewhat post hoc rather than central to the paper as written, and the proposed benchmark remains too underspecified to establish a compelling technical contribution on its own. The methodology still has very limited technical novelty and relatively weak validation of the central constructs. Several of the main claims, especially around Grok’s role as an arbiter, the engagement gap, and shallow alignment, are interesting and potentially important, but they are not established with the level of rigor I would expect for ICML. More broadly, the work still reads more naturally as a computational social science or human-AI interaction study than as a core ML paper advancing machine learning methodology, evaluation, or benchmarking in a sufficiently central way. More natural venues would be ICWSM, CSCW, or FAccT where large-scale analysis of public discourse, platform behavior, and sociotechnical implications are more central.